# Systematic Review of Equine Influenza A Virus Vaccine Studies and Meta-Analysis of Vaccine Efficacy

**DOI:** 10.3390/v15122337

**Published:** 2023-11-28

**Authors:** Sol Elliott, Olaolu T. Olufemi, Janet M. Daly

**Affiliations:** One Virology, Wolfson Centre for Global Virus Research, School of Veterinary Medicine and Science, University of Nottingham, Sutton Bonington LE12 5RD, UK

**Keywords:** equine influenza, vaccine efficacy, vaccination

## Abstract

Vaccines against equine influenza have been available since the late 1960s, but outbreaks continue to occur periodically, affecting both vaccinated and unvaccinated animals. The aim of this study was to systematically evaluate the efficacy of vaccines against influenza A virus in horses (equine IAV). For this, PubMed, CAB abstracts, and Web of Science were searched for controlled trials of equine IAV vaccines published up to December 2020. Forty-three articles reporting equine IAV vaccination and challenge studies in previously naïve equids using an appropriate comparison group were included in a qualitative analysis of vaccine efficacy. A value for vaccine efficacy (VE) was calculated as the percentage reduction in nasopharyngeal virus shedding detected by virus isolation in embryonated hens’ eggs from 38 articles. Among 21 studies involving commercial vaccines, the mean VE was 50.03% (95% CI: 23.35–76.71%), ranging from 0 to 100%. Among 17 studies reporting the use of experimental vaccines, the mean VE was 40.37% (95% CI: 19.64–62.44), and the range was again 0–100%. Overall, complete protection from virus shedding was achieved in five studies. In conclusion, although commercially available vaccines can, in some circumstances, offer complete protection from infection, the requirement for frequent vaccination in the field to limit virus shedding and hence transmission is apparent. Although most studies were conducted by a few centres, a lack of consistent study design made comparisons difficult.

## 1. Introduction

Equine influenza is a major respiratory disease of equids caused by influenza A virus (IAV). In immunologically naïve animals, clinical signs of disease usually appear 2 or more days after infection and typically include elevated body temperature, nasal discharge, and cough. Although rarely fatal, equine IAV is highly contagious and is associated with high morbidity in susceptible animals [1].

Influenza A viruses are classified into subtypes based on the antigenic properties of the two surface glycoproteins—haemagglutinin (HA) and neuraminidase (NA). Two subtypes (H7N7 and H3N8) have been associated with endemic disease in equids, but the last isolation of an H7N7 subtype virus was made in 1989 [2]. However, viruses of the H3N8 subtype have continued to circulate since they were first isolated from horses in North America in 1963 [3].

Inactivated virus vaccines against equine IAV became available shortly after the emergence of the H3N8 subtype [4]. In the UK, vaccine uptake fluctuated with the occurrence of outbreaks (around every 3 years) until 1979, when a major epidemic affected both unvaccinated and vaccinated horses in Europe [5]. This was the first indication that vaccine strains needed to be updated to maintain vaccine effectiveness (Figure 1). This epidemic also led to mandatory vaccination for racehorses in the UK, Ireland, and France. Equine IAV of the H3N8 subtype continued to evolve and caused a further European epidemic in 1989. It was subsequently recognised that two evolutionarily distinct lineages of equine H3N8 viruses were circulating in the Americas and Europe/Asia [6]. After a meeting between World Organisation for Animal Health (WOAH, formerly OIE) and World Health Organization (WHO) experts on equine influenza in 1995, a formal process for annually reviewing the composition of equine IAV vaccines was established [7]. Further evolution of equine IAV H3N8 strains led to the identification of sub-lineages of the American lineage: South American, Kentucky, and Florida [8]. The Florida sub-lineage further split into clades, Florida clade 1, FC1, and FC2 [9], necessitating further updates to vaccine strain recommendations.

There are multiple different types of equine IAV vaccine (reviewed in [10]), with different platforms favoured in different regions. Despite the widespread use of vaccines in some populations of horses, equine H3N8 viruses continue to circulate and periodically cause major outbreaks, most recently in the Americas, Europe, and Africa in 2018–2020 [11,12,13].

We performed a systematic review of controlled clinical trials to assess the efficacy of different equine IAV vaccines. The requirements for efficacy testing of equine IAV vaccines are described in the WOAH manual [14]. Vaccine efficacy is measured by experimental vaccination and challenge studies in the host species (i.e., horses or ponies). The challenge is performed by exposing vaccinated and unvaccinated or placebo-vaccinated control animals to infectious viruses and comparing clinical signs, virus shedding, and serological responses. The single radial haemolysis (SRH) assay is recommended by WOAH for the measurement of antibodies, but the haemagglutination inhibition (HI) test may be used.

## 2. Materials and Methods

The specific question addressed in this systematic review was “what is the efficacy of equine influenza A virus vaccines?”. The PICO elements answered were as follows: Population: equids (horses or ponies); Intervention: equine influenza A virus vaccine; Comparator: placebo vaccine or no vaccination; and Outcomes: nasopharyngeal virus shedding measured and/or seroconversion (a meaningful increase in antibody). In addition to the PICO elements, another inclusion criterion was any controlled vaccination and challenge infection trial in horses or ponies. No restrictions were placed on language.

The identification and screening of the literature were carried out with reference to the PRISMA (Preferred Reporting Items for Systematic Reviews and Meta-Analyses) guidelines [15]. The NIH National Library of Medicine (PubMed.gov), CAB abstracts, and Web of Science core collection were searched for articles published up to December 2020. The following search terms were used to search PubMed, with a similar search structure used for CAB abstracts and Web of Science: (((“influenza A virus” [MeSH Terms] OR “influenza virus” [All Fields]) AND (“equidae” [MeSH Terms] OR “equine” [All Fields])) AND (“vaccines” [MeSH Terms] OR “vacc*” [All Fields])). The results from the searches were downloaded into a bibliographic software program (EndNote X9, Clarivate Analytics, Philadelphia, PA, USA) and de-duplicated.

**Selection process:** There were two stages of screening. First, the titles and abstracts of each article identified by the search strategy were independently assessed by two reviewers (SE and JMD) for relevance using the following primary screening questions: “Does the title and/or abstract describe a primary research study?” and “Does the title and/or abstract describe a vaccine efficacy study conducted in equids?”

The second stage of screening involved two independent reviewers (SE and JMD) assessing the full text of each article deemed eligible by the first stage of screening. The following secondary screening questions were used to assess the full text of each: (i) “Is this a primary research study?” (ii) “Does this article include an equine IAV vaccination and challenge study in previously naïve equids?”, (iii) “Does this article report using an appropriate comparison group?”, and (iv) “Does the article examine one of the following outcomes: seroconversion, virus shedding, clinical signs?”

During screening, a reference was only excluded if both reviewers answered no to any screening question. Any conflicts were resolved by consensus. If consensus could not be reached, the third person on the review team (OTO) was consulted.

**Data extraction:** Two reviewers independently extracted data from eligible studies. The datasheet was pilot-tested to ensure consistency in data extraction. Authors were not contacted to request missing data or to clarify published results. The following information was extracted: (A) study information: year of publication, the purpose of study, study design (randomisation and blinding); (B) population information: breed, age, sex; (C) intervention and comparator information: Intervention: vaccine type (e.g., inactivated whole virus, live attenuated, commercial, or experimental), viruses included in the vaccine, route of administration, number of doses, the interval between doses; comparator: unvaccinated, placebo; (D) challenge and outcomes: interval from last vaccine dose to challenge, virus isolate and dose, route of administration (e.g., intranasal instillation and aerosol to individual or group); method(s) used to identify virus shedding and duration and methods used to measure antibodies. Additional information collected (e.g., adjuvant, duration of virus shedding, reduction/prevention of clinical signs, and virus isolate used to quantify antibody responses) was not used in the qualitative or quantitative synthesis.

**Data analysis:** Vaccine efficacy (VE) was calculated as VE = 1 − (% positive in vaccine group/% positive in placebo group) × 100. Data for Forest plots were generated STATA [16] and plots created using MedCalc^®^ Statistical Software version 22.016 (MedCalc Software Ltd., Ostend, Belgium; https://www.medcalc.org; 2023)

## 3. Results

### 3.1. Qualitative Analysis

The literature search resulted in a total of 1817 records (792 after duplicates were removed), 58 of which were deemed potentially relevant after screening the titles and abstracts (Figure 2). After screening full texts, 43 articles met the inclusion criteria and were included in qualitative analysis. Only five of the articles were published between 1983 and 1998 (Table 1). From 1999 until 2020, between one and four articles were published each year, with the exception that no articles were published in 2002, 2015, and 2017.

#### 3.1.1. Study Purpose and Design

The reported studies were conducted with a variety of aims, including testing the safety and immunogenicity of vaccines under development and the onset and duration of immunity of vaccines under development or vaccines already in commercial use (Table 1). Different vaccination regimes were tested (e.g., single versus two doses and challenge during the ‘immunity gap’), different adjuvants, and combination with other immunogens (e.g., tetanus toxoid) as well as different vaccine delivery sites and methods (e.g., systemic prime and mucosal boost). Some studies were conducted specifically to assess the induction of cell-mediated immunity. The cross-protective efficacy of vaccines or efficacy of vaccines, including updated virus isolates, were tested. Finally, the efficacy of vaccines in specific populations (e.g., older animals or those undergoing rigorous exercise) was tested.

Twenty of the articles did not mention whether animals were randomly allocated to different groups (Table 1). Of those that mentioned randomisation, three stated that the randomised permuted block method was used, one used randomisation based on sex and animal identification number using a four-element permutation, one used an ‘online randomisation generator’, one randomised ‘based on microchip number’, one used the random number generating function in Excel and one used SAS1 v8.2 software. It was only specifically stated in one study that investigators were not blinded; in the majority (28) of the articles, no statement was made about blinding of investigators. In five articles, the use of blinding was reported but without providing any detail. In four articles, it was stated that investigators evaluating clinical signs were blinded, and an additional article stated that the investigator evaluating clinical signs in one study was blinded to the identities of the vaccinates but that this was not possible for the second study. In three articles, it was reported that investigators performing clinical observations and performing laboratory work were blinded, and in one, “double blinding” was used.

#### 3.1.2. Study Population Information

The breed used was not stated in eight articles (Table 2). Of those that reported breed, most used Welsh mountain ponies (n15 articles), three used Norwegian Fjord ponies, two used Shetland ponies, four used “ponies”, two used Kazakh dual-purpose Mugalzhar, five used ‘mixed breed’, and two used various breeds. The age of animals was not stated in seven of the articles. In the majority that provided information, yearlings or 1–2-year-olds were used; the youngest animals were 4–6 months, and the oldest were 20–28 years. The majority (23) of articles did not provide the sex of the animals. In 16 articles, a mix of male and female animals was used (one study specified 10 male and 2 female), and only male animals were used in 4 articles.

#### 3.1.3. Intervention and Comparator Information

Commercially available vaccines were used in 26 (60.5%) of the articles (Table 3). These included the inactivated whole virus vaccines (Duvaxyn IE-T Plus, Equilis prequenza TE, Equilis Resequin), an ISCOM vaccine (Equip-F), a canarypox vectored vaccine (sold as Recombitek and Proteq-Flu), and a ‘modified live’ or ‘live attenuated’ vaccine (Flu-Avert IN). Equilis prequenza TE is described as an inactivated whole virus vaccine or an ISCOM-Matrix/ISCOMatrix vaccine. The type of vaccine was not stated in three articles in which Duvaxyn IE-T Plus, Equilis prequenza TE, or a ‘Fort Dodge vaccine’ was used. In one article, Equilis prequenza TE is described as containing ‘purified antigens’, and in another, Duvaxyn IE plus is described as containing ‘egg-produced antigens’. Studies with experimental vaccines included studies of the commercial vaccines during their development and studies of vaccines that were not subsequently commercialised, for example, the DNA vaccines as well as inactivated virus vaccines containing a single virus with no adjuvant used to test the impact of antigenic drift on vaccine efficacy.

The route of administration varied with the type of vaccine, with most studies (22/43, 52.4%) using intramuscular (IM) injection. There were seven studies in which intranasal inoculation was used, and seven articles did not provide the route of inoculation. The remaining seven studies that used other routes of inoculation included intramuscular followed by intranasal to test a systemic prime/mucosal boost regimen, and different nebuliser devices were used to deliver attenuated viruses. Finally, DNA vaccines were delivered using biolistic devices (‘gene gun’) or other devices such as the PharmaJet^®^ needle-free delivery device.

The vaccines contained a wide range of viruses, including H7N7 subtypes (Prague/56, Cornell/16/74, and Newmarket/77). The H3N8 viruses represented in vaccines span the phylogeny of the virus from 1963 to 2007: Miami/63, Fontainebleau/79, Brentwood/79, Kentucky/81, Suffolk/89, Arundel/91, Borlänge/92, Newmarket/1/93, Newmarket/2/93, Kentucky/94, Kentucky/5/02, Ohio/03, South Africa/4/03, Bari/05, Aboyne/05, Otar/764/07, and Richmond/1/07.

In most studies (34/43 = 79.0%), the control group was left unvaccinated. In the remainder, phosphate-buffered saline (2), Carbomer 974P diluent (1), tetanus toxoid diluent (1), ‘diluent’ (1), ‘sham DNA’ were used, or the treatment of the controls was not stated (3).

#### 3.1.4. Challenge Information and Outcome Measures

In the studies analysed, intranasal instillation was used in 3, nebulisation into a room was used in 22, and individual aerosol delivery was used in 18 (Table 4). Only one study involved a challenge with an H7N7 subtype virus. There were 21 different H3N8 viruses used as challenge strains in the remaining studies, with isolation dates ranging from 1963 to 2014.

Clinical disease was a reported outcome measure in most studies, but the clinical signs noted and scoring systems used were very diverse. Virus isolation in embryonated hens’ eggs was used in most studies. In two studies, the Directigen Flu A test, which detects viral protein, or RT-qPCR, was used instead of virus isolation in eggs, and one study used Madin–Darby canine kidney cells for virus isolation. Ten of the studies, several conducted by the same research group, measured virus shedding by both VI in eggs and RT-qPCR and three of these additionally detected viral nucleoprotein by ELISA.

The SRH assay alone was used in 20 of the 43 studies to measure equine IAV-specific antibody levels, followed by HI only (9 studies). ELISA was used as the sole measure of antibodies in four studies and in combination with SRH (two studies), HI (two studies), or a virus neutralisation test in one study.

Cell-mediated immunity (CMI) was assessed in a subset of studies; two used a tritiated thymidine incorporation assay to measure virus-specific lymphoproliferation, and two measured interferon-gamma synthesising cells, but it was not specified which of the assays described in the cited article were used.

### 3.2. Quantitative Analysis

None of the studies reported a value for vaccine efficacy. Here, we calculated vaccine efficacy as the reduction in the proportion of animals shedding virus (determined by virus isolation in embryonated hens’ eggs) compared to a control group that received no vaccine. Six studies were excluded from the quantitative analysis because they did not report the number of animals shedding virus in each group; used the Directigen FluA test, virus isolation in MDCK cells, or RT-qPCR rather than isolation in embryonated hens’ eggs; or there was an internal discrepancy in the results reported.

Vaccine efficacies calculated for 21 studies in which licensed vaccines were administered are presented as a forest plot in Figure 3. The mean VE was 50.03% (95% CI: 23.35–76.71%), ranging from 0 to 100%. Virus shedding was completely prevented in all vaccinated animals (VE = 100%) in three studies. Among 17 studies reporting the use of experimental vaccines, the mean VE was 40.37% (95% CI: 19.64–62.44), and the range was again 0–100% (Figure 4). Complete protection from virus shedding was achieved in two studies.

## 4. Discussion

This systematic review provides a synthesis of current evidence regarding the efficacy of equine IAV vaccines. Most of the articles reported studies of vaccines under development or experimental application of commercial vaccines (e.g., to determine the impact of age or exercise); studies conducted for licensing purposes may not be published in peer-reviewed journals.

The HI test has been used for decades to determine antibody titres to influenza A viruses. The test determines the highest dilution of serum able to inhibit the ability of haemagglutinin (the receptor-binding protein of IAV) to bind receptors on red blood cells, thus inhibiting agglutination. For diagnosis of infection in the presence of pre-existing antibodies, a 4-fold increase in titre is used to define seroconversion. In contrast, the SRH assay does not involve diluting serum samples and measures complement-mediated lysis of red blood cells coated with the virus. Threshold values of antibodies measured by SRH that afford protection against clinical signs or viral shedding when vaccinated animals are exposed to homologous virus challenge can be defined [60]. Thus, the SRH is the preferred test for measuring vaccine-induced antibodies. Although seroconversion (defined as a 4-fold increase in HI titre as mentioned above or a 2-fold increase or an increase of 50 mm^2^ in SRH zone area in pre- and post-challenge samples) can be used to determine whether animals have been infected, antibody results were most often reported longitudinally to monitor responses to vaccination and challenge rather than as a primary (or secondary) outcome after challenge.

Like the HI test, ELISA only measures the binding capacity of antibodies and is a semi-quantitative method. Virus neutralisation (VN) tests are usually regarded as the gold standard for measuring functional antibody responses. However, these are difficult to perform for influenza A viruses because the virus typically causes limited cytopathic effect, which means that an additional assay (e.g., ELISA or RT-qPCR) has to be performed to measure viral replication. Hence, VN tests were only used in three of the studies: in conjunction with ELISA, HI, or both SRH and HI. More recently, the use of pseudotyped viruses, which package a reporter gene that provides a convenient read-out, have been developed for a wide range of viruses, and their use to measure neutralising antibody responses has gained wider acceptance during the COVID-19 pandemic [61]. The potential application of a pseudotyped virus neutralisation test to measure antibody responses in equine influenza vaccine efficacy studies has been described [62]. This has yet to be fully characterised to determine whether it can provide a correlate of protection or to define seroconversion.

The methods used for assessing equine influenza vaccine efficacy have evolved over time. Initially, experimental infection was achieved by intranasal instillation of the virus. However, it was demonstrated early on that infection with an aerosol of virus led to clinical signs that more closely mimicked natural infection [63]. Initially, the virus was aerosolised using a nebuliser to introduce the virus into a room or enclosed space in which the animals were held as a group for a period. More recently, the infection method has been further refined by using individual masks to deliver aerosolised virus. Garrett et al. (2017) showed that the use of an individual face mask reduced the heterogeneity of clinical responses and virus shedding, thus increasing the statistical power of a study [64]. In the studies analysed, intranasal instillation was only used in three early studies (published in 1983, 1988 and 1999). Group and individual aerosol delivery were used in 22 and 18 studies, respectively.

Although all studies reported the impact of vaccination on clinical disease, the variability in the clinical signs recorded, the subjective nature of many of these and how clinical scores were defined meant that comparison of clinical disease as an outcome between studies was not possible.

Therefore, to compare vaccine efficacy across the reported studies, the proportion of animals shedding virus as determined by virus isolation in eggs (the method most consistently performed across the studies) was used. Only three studies used alternative methods (the Directigen Flu A test, which detects viral protein, RT-qPCR, or virus isolation on Madin–Darby canine kidney cells). Ten of the studies, several of which were conducted by the same research group, measured virus shedding by both virus isolation in eggs and RT-qPCR; these studies consistently showed that RT-qPCR was the more sensitive technique. However, the biological relevance of detecting traces of viral RNA, which may not indicate the presence of an infectious virus, is called into question.

This assessment of VE is very stringent as the threshold of antibodies required to suppress virus shedding is much higher than for protection against clinical disease (SRH antibody levels ≥ 150 mm^2^ versus 85 mm^2^) [60]. Nonetheless, complete prevention of viral shedding (100% VE) was achieved in three studies of commercial vaccines. This included one group that received three doses of vaccine [41]. In the other two studies, a canarypox-vectored vaccine [42] and an ISCOM [43] vaccine containing different virus strains were tested under similar conditions (two doses given around 6 weeks apart) by exposure to infectious virus (Newmarket/5/03 and South Africa/4/03, respectively) 2 weeks after the second dose.

It is difficult to draw inferences on the relative efficacy of different commercially available vaccines as most studies differed in more than one aspect. Only two of the published studies directly compared vaccines under the same conditions. In the more recent of these, horses were challenged by individual aerosol with A/equine/Wexford/14 (H3N8) 120 days after the second dose of vaccine [35]. The VE of the ISCOM vaccine containing the H3N8 strains A/equine/Newmarket/2/93 and A/equine/Richmond/1/07 was only 14%. However, the canarypox-vectored vaccine containing A/equine/Ohio/03 and A/equine/Richmond/07 failed to completely prevent virus shedding in any of the vaccinated animals (VE = 0%). The relatively long interval before the challenge (almost 4 months) could account for the poor VE, although this was also the only study in which A/equine/Wexford/14 (H3N8) was used for the challenge.

The other study [21] that directly compared two vaccines compared the canarypox-vectored vaccine containing A/equine/Newmarket/2/93 (H3N8) and A/equine/Kentucky/94 (H3N8) and an ISCOM vaccine containing A/equine/Newmarket/77 (H7N7), A/equine/Borlänge/91 (H3N8), and A/equine/Kentucky/98 (H3N8). The ponies were challenged individually by exposure to aerosol with A/equine/Sydney/07 (H3N8) 2 weeks after the second vaccine dose. The VE for the canarypox-vectored vaccine was 20%, while for the ISCOM vaccine, it was 60%. The inclusion of different virus strains in the two vaccines might have contributed to the differing VE values obtained; the authors noted that the composition of the lower efficacy canarypox-vectored vaccine was updated shortly after the study had been performed [21]. Even though most of the studies involving commercial vaccines were designed to study cross-protection against heterologous challenge viruses or test vaccines with updated strains, it is difficult to assess the extent to which a vaccine ‘mismatch’ with the challenge virus affects vaccine efficacy. Two studies [26,59] used non-commercial, unadjuvanted monovalent vaccines to demonstrate the impact of challenge with a heterologous strain (Figure 4). Not all reports that used commercially available vaccines detailed the composition of the vaccines at the time of the study. Two studies [48,50] in which the same commercial vaccine was tested under the same conditions but with different challenge viruses (A/equine/Ohio/03 [H3N8] and A/equine/Richmond/07 [H3N8]) gave the same VE value (43%). One article reported two studies in which a single dose of Flu Avert IN containing the American-lineage A/equine/Kentucky/91 (H3N8) as the only vaccine strain was administered to similar-aged horses with challenge infection 4 weeks later with two different virus isolates [22]. The VE was 0% when challenged with American-lineage A/equine/Kentucky/98 (H3N8) and 50% when challenged with the European-lineage virus A/equine/Saskatoon/90 (H3N8), isolated from a quarantined horse in Canada. However, this comparison is confounded by the use of an individual nebuliser for challenge with the Kentucky/98 virus and the exposure of the group to an aerosol of the Saskatoon/90 virus. For studies using aerosolisation of the virus in a room, the mean VE was 56%, whereas for studies using individual aerosol delivery, it was 32%.

Some of the articles described using commercial vaccines to study how host factors influence the response to vaccination. For example, Adams et al. (2011) showed that VE was slightly higher in older animals (76%) than in younger naïve animals (71%) [17]. Lunn et al. (2001) demonstrated that vaccination of ponies after 5 days of strenuous exercise on a high-speed treadmill resulting in immunosuppression reduced the efficacy of vaccination when ponies were challenged 3 months later; all the ponies in the exercised group shed virus (VE = 0%) compared to 25% VE in an unexercised group [36].

Overall, the variation in study design meant that it was not possible to compare results for different vaccines across studies. It would also be difficult to extrapolate from the studies described in this review to the field situation. The WOAH manual suggests that the challenge should be carried out no fewer than 2 weeks and preferably more than 3 months after the second dose of vaccine. However, longer-duration studies are more costly, and most of the studies included groups of animals that were challenged at 2 (n = 15) or 4 (n = 13) weeks after a second dose of vaccine. In three of the five studies that gave a VE of 100%, the challenge was 2 weeks after the second dose. Thus, it could be argued that most studies used a schedule that presented a ‘best-case’ scenario. On the other hand, all animals, except one group of older horses in Adams et al. [17], were naïve at the start of the study (one of the inclusion criteria), and most only received one or two doses of vaccine. In the study by Mumford et al. (1994), 100% VE was seen in the group that received two doses of ISCOM vaccine 6 weeks apart with a booster dose 5 months later and challenge 15 months after the third dose [41]. This would appear to support recommendations for more frequent than annual vaccination, at least for younger animals at high risk of exposure. Furthermore, the amount of virus shed was not taken into consideration when determining the VE. The amount of virus shed may have been reduced sufficiently to prevent transmission, thus contributing to herd immunity, and it is likely that clinical disease was suppressed even when the VE was relatively low, providing benefit to vaccinated individuals.

## Figures and Tables

**Figure 1 viruses-15-02337-f001:**
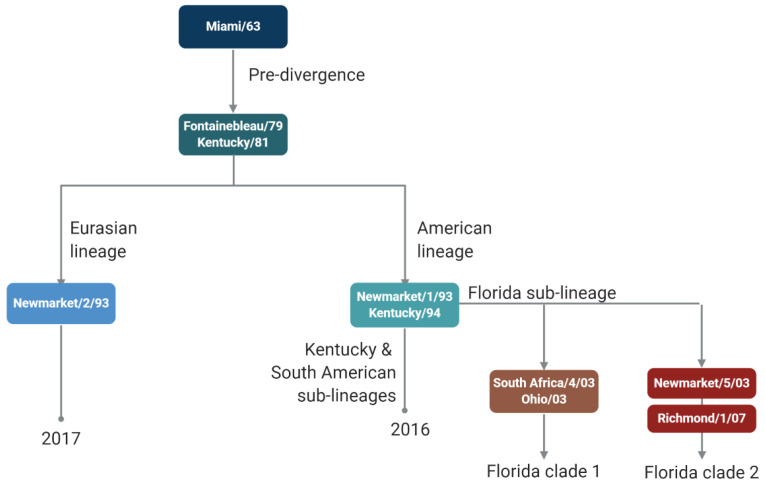
Schematic of equine influenza A virus vaccine strain recommendations.

**Figure 2 viruses-15-02337-f002:**
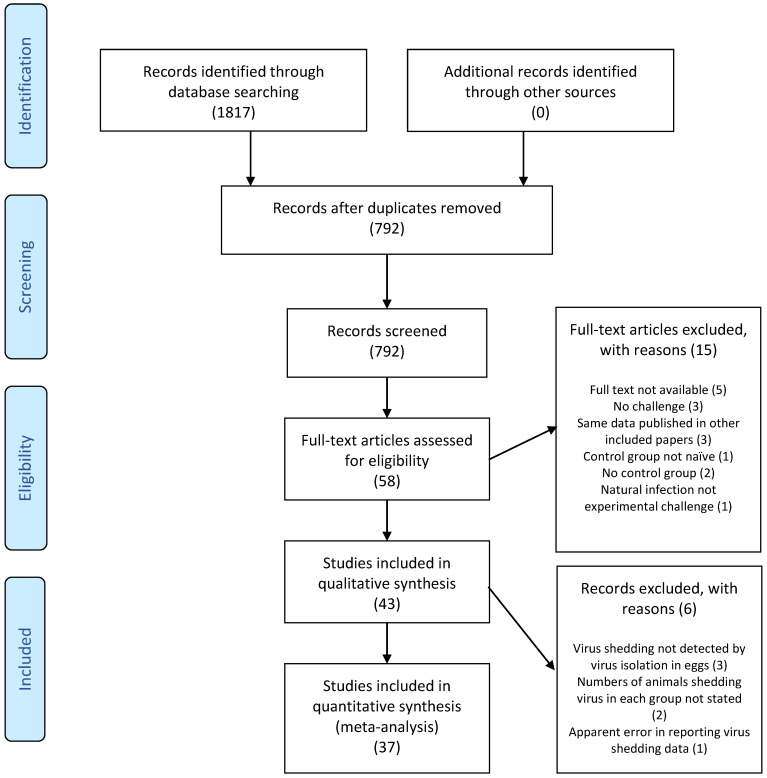
PRISMA flow diagram [15]. Numerals in brackets indicate the number of articles.

**Figure 3 viruses-15-02337-f003:**
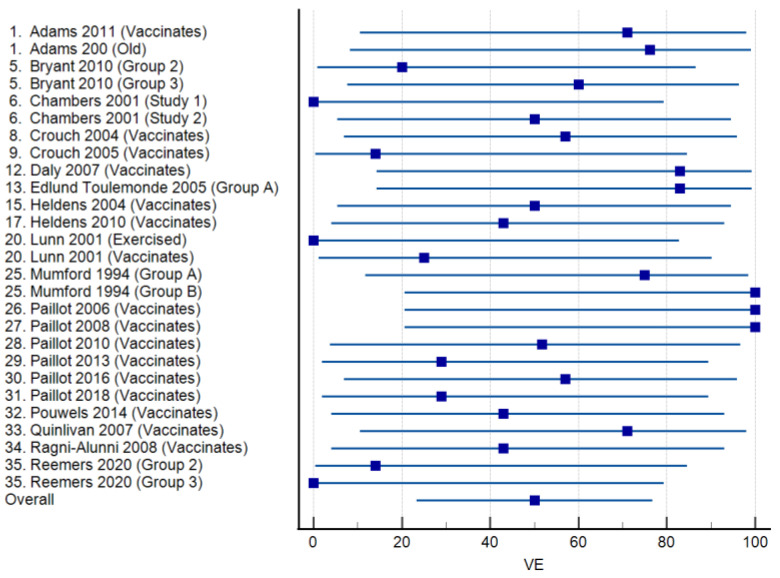
Forest plots of vaccine efficacy for 21 studies [17,21,22,24,25,28,29,31,33,36,41,42,43,44,45,46,47,48,49,50,51] involving commercially available equine influenza A vaccines. The squares show the percent vaccine efficacy (VE) for each study with the 95% confidence intervals indicated by horizontal lines.

**Figure 4 viruses-15-02337-f004:**
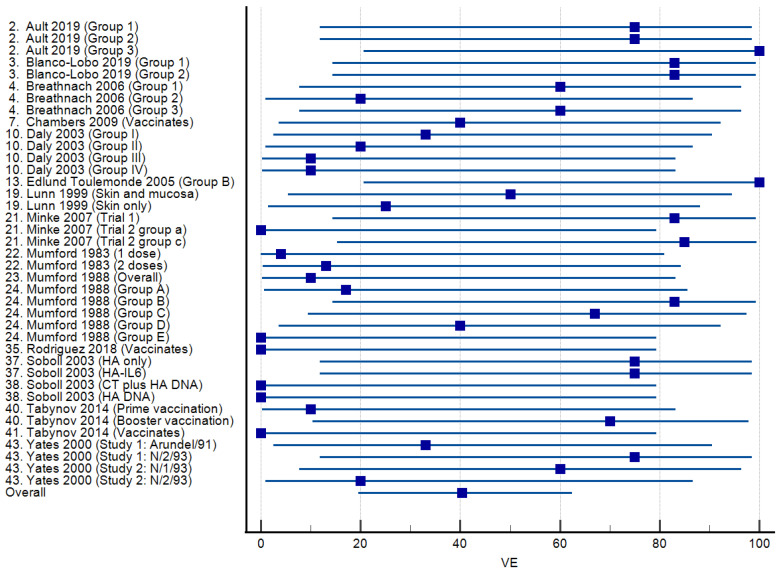
Forest plots of vaccine efficacy for 17 studies [18,19,20,23,26,29,35,37,38,39,40,51,53,54,56,57,59] involving experimental equine influenza A vaccines. The squares show the percent vaccine efficacy (VE) for each study with the 95% confidence intervals indicated by horizontal lines.

**Table 1 viruses-15-02337-t001:** Information on design of 43 studies included in the qualitative analysis.

No.	First Author	Year	Purpose of Study	Randomisation	Blinding
1	Adams	2011 [17]	Old vs. naïve	yes	ns
2	Ault	2012 [18]	Compare delivery methods	yes	yes ^7^
3	Blanco-Lobo	2019 [19]	Test efficacy with updated vaccine strain	yes	yes ^7^
4	Breathnach	2006 [20]	Compare rMVA vaccination with a DNA priming dose and nucleoprotein (NP) versus haemagglutinin (HA) vaccination	ns	ns
5	Bryant	2010 [21]	Compare efficacy of two commercial vaccines	yes	ns
6	Chambers	2001 [22]	Heterologous challenge	yes	yes ^8^
7	Chambers	2009 [23]	Compare three modified live vaccines	yes	ns
8	Crouch	2004 [24]	Test efficacy with updated vaccine strain	yes	ns
9	Crouch	2005 [25]	Test systemic prime/mucosal boost regimen	yes ^1^	ns
10	Daly	2003 [26]	Test cross-protective efficacy	yes	No
11	Daly	2004 [27]	Test cross-protective efficacy	ns	ns
12	Daly	2007 [28]	Test cross-protective efficacy	yes ^1^	yes
13	Edlund Toulemonde	2005 [29]	Compare single versus two doses	yes ^1^	ns
14	Folsom	2001 [30]	Efficacy after two doses and impact of exercise	ns	ns
15	Heldens	2004 [31]	Duration of immunity	yes	ns
16	Heldens	2009 [32]	Onset and duration of immunity	ns	ns
17	Heldens	2010 [33]	Duration of immunity	yes	yes ^7^
18	Holmes	1988 [34]	Test efficacy	ns	yes
19	Lunn	1999 [35]	Compare vaccination sites	ns	ns
20	Lunn	2001 [36]	Test impact of exercise	ns	ns
21	Minke	2007 [37]	Efficacy of a new vaccine	yes	ns
22	Mumford	1983 [38]	Compare graded doses of vaccine	ns	ns
23	Mumford	1988 [39]	Examine relationship between vaccine-induced antibodies and protective efficacy	ns	ns
24	Mumford	1994 [40]	Compare efficacy of different adjuvants	ns	ns
25	Mumford	1994 [41]	Compare two doses with tetanus toxoid and booster without or with three doses without	ns	ns
26	Paillot	2006 [42]	Measure cell-mediated immunity	ns	ns
27	Paillot	2008 [43]	Measure cell-mediated immunity	yes	ns
28	Paillot	2010 [44]	Test cross-protective efficacy	yes	yes
29	Paillot	2013 [45]	Test cross-protective efficacy	ns	yes
30	Paillot	2016 [46]	Test efficacy with updated strain at ‘minimum protective dose’	yes ^2^	yes ^9^
31	Paillot	2018 [47]	Test efficacy when challenged in ‘immunity gap’	yes ^3^	yes ^9^
32	Pouwels	2014 [48]	Test cross-protective efficacy	yes ^4^	yes ^7^
33	Quinlivan	2007 [49]	Measure pro-inflammatory and antiviral cytokine expression	yes ^5^	ns
34	Ragni-Alunni	2008 [50]	Test cross-protective efficacy	ns	ns
35	Reemers	2020 [51]	Compare cross-protective efficacy of two commercial vaccines	yes	yes ^7^
36	Rodriguez	2018 [52]	Test efficacy of a novel vaccine	ns	ns
37	Soboll	2003 [53]	Antibody and cellular immune responses to a DNA vaccine	ns	ns
38	Soboll	2003 [54]	Evaluate cholera toxin as an adjuvant for a DNA vaccine	ns	ns
39	Soboll	2010 [55]	Onset and duration of immunity to a commercial vaccine	yes ^6^	yes
40	Tabynov	2014 [56]	Safety and immunogenicity of a novel cold-adapted modified live virus vaccine	ns	ns
41	Tabynov	2014 [57]	Duration of immunity to a novel cold-adapted modified live virus vaccine	ns	ns
42	Townsend	2001 [58]	Efficacy of a cold-adapted intranasal vaccine	yes	yes ^10^
43	Yates	2000 [59]	Test cross-protective efficacy	ns	ns

^1^ Randomised permuted block method; ^2^ randomisation based on sex and identification number using a 4-element permutation table; ^3^ online randomisation generator; ^4^ based on microchip number; ^5^ random number generating function in Microsoft Excel; ^6^ SAS1 v8.2 software; ^7^ clinical observations; ^8^ in the Saskatoon/90 trial, the investigator evaluating clinical signs was blinded to the identities of the vaccinates. This was not possible in the Kentucky/98 trial; ^9^ clinical observations and laboratory work; ^10^ double blind. ns = not stated.

**Table 2 viruses-15-02337-t002:** Study population information from 43 studies included in the qualitative analysis.

No.	Total No. Animals	Breed	Age	Sex
1	28	mixed-light breeds	old horses: 20–28 years; young horses: 7–10 months old	not stated
2	16	Shetland blood, Welsh blood, and Florida swamp pony blood	1–2 years	male and female
3	18	not stated	1–2 years	male and female
4	20	not stated	yearlings	not stated
5	15 ^1^	Welsh mountain pony	1–2 years	not stated
6	28	not stated	7 months	not stated
7	9 then 8 ^2^	not stated	yearlings	not stated
8	14	Welsh mountain pony	not stated	male and female
9	14	Welsh mountain pony	not stated	male and female
10	50 ^3^	Welsh mountain pony	not stated	not stated
11	60	not stated	not stated	not stated
12	14	Welsh mountain pony	approx. 11 months	male
13	15	Welsh mountain pony	1 year	male
14	12	mixed-breed ponies	not stated	not stated
15	11	Fjord	6 months	not stated
16	24	Fjord	4–7 months	not stated
17	12	Fjord	4–7 months	not stated
18	51	Mixed breed pony (Welsh mountain type)	yearlings and 2 years	not stated
19	12	not stated	1–7 years	male and female
20	15	pony	9–15 months	male and female
21	49	Welsh mountain pony	1–3 years	male
22	46	Welsh mountain pony	yearlings	not stated
23	31	Welsh mountain pony	yearlings	not stated
24	29	not stated	not stated	not stated
25	35	not stated	4–6 months	not stated
26	24	Welsh mountain pony	9 months	not stated
27	10	Welsh mountain pony	yearlings	not stated
28	12	Welsh mountain pony	12 months	not stated
29	12	Welsh mountain pony	6–8 months	not stated
30	14	Welsh mountain pony	10 months	male and female
31	12	Welsh mountain pony	11 months	male (10) and female (2)
32	12	Shetland pony	2–17 years	not stated
33	14	mixed	5–7 months	male and female
34	13	Shetland pony	10–17 months	not stated
35	19	Norwegian Fjord horse	4–4.5 years	male and female
36	6	mixed breed (mainly Standardbred-quarter horse cross)	1–2 years	male and female
37	25	ponies	1–6 years	male and female
38	12	ponies	1-year-olds	male and female
39	23 (duration)	ponies	6 months	male
	20 (onset)			
40	30	Kazakh dual-purpose Mugalzhar	1–1.5 years	male and female
41	16	Kazakh dual-purpose Mugalzhar	1–1.5 years	male and female
42	90 (29 in challenge study)	Belgian, Percheron, Percheron-Clydesdale cross, and Quarter Horse cross	11 months	male and female
43	30 (study 1) 28 (study 2)	Welsh mountain pony	not stated	not stated

^1^ The study included an additional 4 ponies that were experimentally infected with A/equine/South Africa/4/03 (H3N8) 18 months prior to the study; ^2^ one of the six vaccinates (chosen at random) was omitted because of lack of space; ^3^ one animal in group 1 could not be swabbed safely.

**Table 3 viruses-15-02337-t003:** Intervention and comparator information from 43 studies included in the qualitative analysis.

No	Groups	No. Doses	Interval between Doses	Vaccine	Vaccine Type (Administration Route)	Vaccine Composition A/Equine/^10^	Control
1	Old	1	5 weeks	RECOMBITEK^®^	Canarypox (IM)	Newmarket/2/93, Kentucky/94	Diluent
	Naive	2					
2	Group 1	3	4 weeks	Experimental	DNA (other ^3^)	Ohio/03	Sham DNA
	Group 2					Ohio/03	
	Group 3					Ohio/03, Bari/05, Aboyne/05	
3	Group 1 and group 2	2	29 days	Experimental	LAV (other ^4^)	Ohio/03, Richmond/07	Unvaccinated
4	Group 1	3	42 days, 28 days	Experimental	DNA/MVA (other ^5^)	Kentucky/1/81	Unvaccinated
	Group 2				DNA/MVA (other ^5^)		
	Group 3				MVA (other ^5^)		
5	Group 2	2	5 weeks	Proteq-Flu	Canarypox (IM)	Newmarket/2/93, Kentucky/94	Unvaccinated
	Group 3			Equip-F	ISCOM (IM)	Newmarket/77 (H7N7), Borlänge/91, Kentucky/98	
6	Study 1	1	N/A	FluAvert IN	LAV (IN)	Kentucky/91	“Seronegative”
	Study 2						
7	Group 1	2	4 weeks	Experimental	MLV (IN)	Kentucky/5/02	Unvaccinated
	Group 2						
	Group 3						
8	Group 1	2	6 weeks	Equip F	ISCOM (IM)	Newmarket/77 (H7N7), Borlänge/91 and Kentucky/98	Unvaccinated
9	Group 1	2	6 weeks	Equip F	ISCOM (IM–IN)	Newmarket/77 (H7N7), Borlänge/91 and Kentucky/98	Unvaccinated
10	Group I	2	4 weeks	Experimental	Inactivated virus-no adjuvant (IM)	Suffolk/89	Unvaccinated
	Group II					Kentucky/81	
	Group III					Fontainebleau/79	
	Group IV					Miami/63	
11	Study 1: N/1/93	2	4 weeks	Experimental	Inactivated virus-no adjuvant (IM)	Newmarket/1/93	Unvaccinated
	Study 1: N/2/93					Newmarket/2/93	
	Study 2: N/1/93					Newmarket/1/93	
	Study 2: N/2/93					Newmarket/2/93	
12	Group 1	2	28 days	Duvaxyn IE-T Plus	Not stated ^6^ (IM)	Prague/56 (H7N7), Suffolk/89, Newmarket/1/93	Unvaccinated
13	Group 1	1	5 weeks	Proteq Flu	Canarypox (IM)	Newmarket/2/93, Kentucky/94	Unvaccinated
	Group 2	2					
14	Rested	2	not stated	Fort Dodge vaccine	Not stated ^6^	Miami/63	Unvaccinated
	Exercised						
15	Group 1	2	4 weeks	Equilis resequin	Inactivated whole virus (IM)	Prague/56 (H7N7), Newmarket/1/93 and Newmarket/2/93	Unvaccinated
16	Group 1	2	4 weeks	Equilis prequenza	ISCOM-Matrix (IM)	Prague/56 (H7N7), Newmarket/1/93 and Newmarket/2/93	Unvaccinated
17	Prequenza Te	3	4 weeks 22 weeks	Equilis Prequenza	Subunit vaccine (IM)	Prague/56 (H7N7), Newmarket/1/93 and Newmarket/2/93	Unvaccinated
18	Vaccinees	1 (n = 35)	4 weeks	Experimental	Ts reassortant (not stated)	Cornell/16/74 (H7N7)	Unvaccinated
		2 (n = 4)					
		3 (n = 2)					
19	Skin and mucosa vaccination	3	around 63 days	Experimental	DNA (skin ^7^)	Kentucky/1/81	Unvaccinated
	Skin vaccination						
20	VE (exercised)	1	n/a	FluAvert	MLV (nebuliser ^8^)	Kentucky/1/91	Unvaccinated
	V0 (not exercised)						
21	Trial 1 vaccinates	2	5 weeks	Experimental	Canarypox (not stated)	Newmarket/2/93 Kentucky/94	Tetanus toxoid diluent
	Trial 2 group a	2	5 weeks				
	Trial 2 group c	3	5 weeks 5 months				
22	1 dose	1 ^1^	4 weeks	Experimental	Inactivated whole virus (not stated)	Prague/56 (H7N7), Miami/63	Unvaccinated
	2 doses	2 ^1^					
23	1 dose	1	n/a	Experimental	Inactivated whole virus (not stated)	Miami/63	Unvaccinated
	2 doses	2	4 weeks				
	3 doses	3	4 weeks 10 weeks				
24	Group A: AlPO4 + tetanus combined	3	4 weeks 27 weeks	Experimental	Inactivated whole virus (IM)	Prague/56 (H7N7) Miami/63 Kentucky/81	Unvaccinated
	Group B: Carbomer						
	Group C: Carbomer + tetanus separate sites						
	Group D: Carbomer					Prague/56 (H7N7) Kentucky/81	
	Group E: Carbomer + tetanus combined	2	4.5 weeks			Prague/56 (H7N7) Miami/63 Kentucky/81	
25	Group A (2 doses Equip FT, booster Equip F	3	6 weeks 5 months	Equip F/FT	ISCOM (not stated)	Newmarket/79 (H7N7) Brentwood/79 *	Unvaccinated
	Group B (3 doses Equip F						
26	Vaccinates	2	36 days	ProteqFlu	Canarypox (IM)	Kentucky/94 and Newmarket/2/93	Carbomer 974P diluent
27	Vaccinates	2	6 weeks	Equip F	ISCOM (not stated)	Newmarket/77 (H7N7), Borlänge/91, Kentucky/98	Unvaccinated
28	Group A	2	4 weeks	Duvaxyn IE-T plus	Inactivated whole virus (IM)	Prague/56 (H7N7), Suffolk/89, Newmarket/1/93	Unvaccinated
29	Group A	2	4 weeks	Duvaxyn IE-T Plus	Inactivated whole virus (IM)	Prague/56 (H7N7), Suffolk/89, Newmarket/1/93	Unvaccinated
30	Vaccinates	2	5 weeks	ProteqFlu ^2^	Canarypox (IM)	Ohio/03 Richmond/1/07	Unvaccinated
31	Vaccinates	2	4 weeks	Equilis prequenza TE	Inactivated whole virus (IM)	South Africa/4/03 and Newmarket/2/93	Phosphate buffered saline
32	Vaccinates	2	4 weeks	Equilis prequenza TE	‘purified antigens’ (IM)	Prague/56 (H7N7), Newmarket/1/93 and Newmarket/2/93	Unvaccinated
33	Vaccinates	2	4 weeks	Duvaxyn IE plus	‘egg-produced antigens’ (IM)	Prague/56 (H7N7), Suffolk/89 and Newmarket/1/93	Unvaccinated
34	Vaccinates	2	4 weeks	Equilis prequenza TE	Not stated (IM)	Prague/56 (H7N7), Newmarket/1/93 and Newmarket/2/93	Unvaccinated
35	Group 2	2	4 weeks	Equilis prequenza	Inactivated virus ISCOMatrix (IM)	Newmarket/2/93, South Africa/4/03	Unvaccinated
	Group 3			ProteqFlu	Canarypox vector (IM)	Ohio/03, Richmond/07	
36	Vaccinates	1	n/a	Experimental	LAIV-ts (IN)	Ohio/1/2003	Unvaccinated
37	HA only	3	70 days 6 weeks	Experimental	DNA (other ^9^)	Kentucky/1/81	Unvaccinated
	HA-IL6						
38	CT plus HA DNA	4	Intranasal instillation D0 and D33 PowderJect XR research device D77 and D113	Experimental	DNA (IN and other ^9^)	Kentucky/1/81	Unvaccinated
	HA DNA						
39	Study 1	2	35 days	Recombitek	Canarypox vector (IM)	Kentucky/94 and Newmarket/2/93	Unvaccinated
	Study 2	1	n/a				
40	Single vaccination	1	n/a	Experimental	LAV-ca (IN)	Otar/764/2007	Phosphate buffered saline
	Double vaccination	2	42 days				
41	Vaccinates	1	n/a	Experimental	LAV-ca (IN)	Otar/764/2007	“Control”
42	Vaccinates	1	N/A	Experimental	LAV-ca (IN)	Kentucky/1/91	“Control”
43	Study 1: Arundel/91	2	4 weeks	Experimental	Monovalent whole virus inactivated without adjuvant (IM)	Arundel/91	Unvaccinated
	Study 1: Newmarket/2/93					Newmarket/2/93	
	Study 2: Newmarket/1/93					Newmarket/1/93	
	Study 2: Newmarket/2/93					Newmarket/2/93	

^1^ “During the month prior to challenge one pony from each sub-group [= different potency of vaccines] was given an additional dose of aqueous vaccine containing sufficient antigen to boost antibody titres and ensure that some individuals in the group had high levels of antibody at the time of challenge”; ^2^ vaccine used at ‘minimum protective dose’ (1/100th of commercial dose); ^3^ IM injection (Group 1) or needle-free delivery system (PharmaJet^®^, PharmaJet, Inc., Golden, CO, USA) using spring-powered jet technology to effectively deliver vaccines sub-dermally (Groups 2 and 3); ^4^ Flexineb II portable equine nebulizer/facemask; ^5^ skin (inguinal and perineal areas) and mucosal (conjunctiva and ventrum of tongue) sites of each pony; ^6^ likely to be inactivated whole virus vaccine; ^7^ PowderJect-XR gene gun; ^8^ disposable nebuliser unit (Salter Labs, Arvin, CA, USA); ^9^ PowderJect-XR1 research device 24× on inguinal skin, 8× on perineal skin, 24× on the ventral tongue and 4× on the conjunctiva and third eyelid; ^10^ H3N8 unless otherwise stated (* presumed typographical error in study 25, which gives subtype as H3N3). Abbreviations: IM, intramuscular; IN, intranasal; LAIV-ts, live attenuated influenza virus—temperature sensitive; LAV-ca, live attenuated virus—cold adapted; MLV, modified live vaccine; MVA, modified vaccinia Ankara; n/a, not applicable.

**Table 4 viruses-15-02337-t004:** Challenge information and outcome measures of 43 studies included in the qualitative analysis.

No.	Interval to Challenge ^1^	Virus ^3^	Method	Outcome Measures	
				Virus Shedding	Antibody
1	15 days	Kentucky/5/02	Nebulised aerosol (room)	VI (eggs)	HI
2	7 weeks	Ohio/03	Nebulised aerosol (room)	VI (eggs), RT-qPCR	SRH and HI
3	4 weeks	Kentucky/14 or Richmond/07	Individual aerosol	VI (eggs)	HI
4	30 days	Kentucky/1/81	Individual aerosol	VI (eggs)	ELISA
5	2 weeks	Sydney/2888-8/07	Individual aerosol	VI (eggs), NP-ELISA, RT-qPCR	SRH
6	4 weeks	Kentucky/98	Individual aerosol	VI (eggs)	HI
		Saskatoon/90	Nebulised aerosol (room)		
7	4 weeks	Kentucky/5/2002	Nebulised aerosol (room)	VI (eggs), RT-qPCR	SRH
8	4 weeks	Newmarket/1/93	Nebulised aerosol (room)	VI (eggs)	SRH and ELISA ^4^
9	4 weeks	Newmarket/1/93	Nebulised aerosol (room)	VI (eggs)	SRH and ELISA ^4^
10	2 weeks	Sussex/89	Nebulised aerosol (room)	VI (eggs)	SRH
11	2 weeks	Newmarket/1/93 or Newmarket/2/93	Nebulised aerosol (room)	VI (eggs)	SRH
12	2 weeks	South Africa/4/03	Nebulised aerosol (room)	VI (eggs)	SRH
13	2 weeks	Newmarket/5/03	Nebulised aerosol (room)	VI (eggs)	SRH
14	6 weeks + 5 days ^2^	Miami/63	Individual aerosol	Directigen test kit	VN and ELISA ^4^
15	4 weeks	Kentucky/95	Individual aerosol	VI (eggs)	SRH
16	4 weeks (onset) 22 weeks (duration)	Kentucky/9/95	Individual aerosol	VI (eggs)	HI
17	54 weeks	Kentucky/95	Individual aerosol	VI (eggs)	HI
18	4 weeks	Cornell/16/74 (H7N7)	Individual aerosol	VI (MDCKs)	HI
19	30 days	Kentucky/1/81	Intranasal instillation	VI (eggs)	HI and ELISA ^4^
20	98 days	Kentucky/91	Intranasal aerosol	VI (eggs)	HI
21	2 weeks (trial 1); 5 months (trial 2 A and B); 12 months (trial 2 C and D)	Sussex/89	Nebulised aerosol (room)	VI (eggs)	SRH
22	22 weeks (single-dose group) 18 weeks (two-dose group)	Newmarket/79	Intranasal instillation	VI (eggs)	SRH and HI
23	13.5 weeks (2-dose group); 3.5 weeks 3-dose group)	Miami/63	Intranasal instillation	VI (eggs)	SRH
24	Groups A–D = 19 weeks; Group E = 18 weeks	Newmarket/79	Nebulised aerosol (room)	VI (eggs)	SRH
25	15 months	Sussex/89	Nebulised aerosol (room)	VI (eggs)	SRH
26	14 days after V2	Newmarket/5/03	Nebulised aerosol (room)	VI (eggs)	SRH
27	2 weeks	South Africa/4/03	Nebulised aerosol (room)	VI (eggs)	SRH
28	2 weeks	Sydney/07	Nebulised aerosol (room)	VI (eggs), NP-ELISA, RT-qPCR	SRH
29	2 weeks	Richmond/1/07	Nebulised aerosol (room)	VI (eggs), NP ELISA, RT-qPCR	SRH
30	2 weeks	Richmond/1/07	Nebulised aerosol (room)	VI (eggs) and RT-qPCR	SRH
31	158 days	Northamptonshire/1/13	Individual aerosol	VI (eggs) and RT-qPCR	SRH and HI
32	3 weeks	Richmond/1/07	Individual aerosol	VI (eggs)	HI, VN (eggs)
33	16 days	Kildare/89	Nebulised aerosol (room)	VI (eggs) and RT-qPCR	SRH
34	3 weeks	Ohio/03	Individual aerosol	VI (eggs)	HI
35	120 days	Wexford/14	Individual aerosol	VI (eggs) and RT-qPCR	SRH, HI and VN (eggs)
36	27 days	Kentucky/1/81	Nebulised aerosol (room)	VI (eggs) and RT-qPCR	HI
37	47 days	Kentucky/1/81	Individual aerosol	VI (eggs)	ELISA
38	81 days	Kentucky/1/81	Individual aerosol	VI (eggs)	ELISA
39	6 months (experiment 1)	Kentucky/91	Individual aerosol	RT-qPCR	ELISA
	14 days (experiment 2)	Ohio/03			
40	12 months	Sydney/2888-8/07	Individual aerosol	VI (eggs)	HI
	1, 2, 3, 4, 5, 6, 9, 12 months	Otar/764/07			
41	28 days	Otar/764/07	Individual aerosol	VI (eggs)	HI and ELISA
42	5 weeks, 6 and 12 months	Kentucky/91	Nebulised aerosol (room)	VI (eggs)	SRH
43	2 weeks	Newmarket/2/93	Nebulised aerosol (room)	VI (eggs)	SRH

^1^ After last vaccine dose; ^2^ vaccinates received 2 doses 6 weeks prior to start of study and were exercised or rested for 5 days before challenge; ^3^ H3N8 subtype unless otherwise stated; ^4^ ELISA used to measure different immunoglobulin G sub-isotypes. Abbreviations: (NP-) ELISA, (nucleoprotein) enzyme-linked immunosorbent assay; HI, haemagglutination inhibition; RT-qPCR, reverse-transcription–quantitative polymerase chain reaction; SRH, single radial haemolysis; VI, virus isolation; VN, virus neutralisation.

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
