# Peer review of "Systematic Review of Equine Influenza A Virus Vaccine Studies and Meta-Analysis of Vaccine Efficacy"

_viruses, 2023, doi:10.3390/v15122337_

Round 1
Reviewer 1 Report
Comments and Suggestions for Authors
A major undertaking, with many difficulties in making a solid metanalysis. The authors present as thorough of an analysis as I believe possible.
Author Response
Thank you for the recognition of the effort this review took!
Reviewer 2 Report
Comments and Suggestions for Authors
The reviewed manuscript presents a meta-analysis of vaccination studies concerning equine influenza vaccines. The manuscript is well-written and provides a useful overview of the details of published vaccine studies that were included based on clearly defined criteria. However, it is not clear why studies without a clearly stated control group (table 3, studies 6, 41, 42) were included in the analysis when the inclusion of an appropriate comparison group was one of the stated inclusion criteria. This should be addressed.
The authors conclude that inconsistent study design did not allow for comparison of the efficacy of different vaccines, but that there was evidence for overall vaccine efficacy in preventing viral shedding. In addition, the authors showed that commercially available vaccines can, in some circumstances, offer complete protection from infection. The stated conclusion that the study offers evidence for frequent vaccination in the field seems a bit less supported – as only few studies addressed this issue – but the presentation of data allows the reader to draw their own conclusions.
The data presentation in tables and figures is appropriate and easy to understand; however, the list of abbreviations in table 3 is not complete (e.g. LAV-ca is not explained, MVA is not explained). This should be corrected.
Author Response
Thank you for highlighting the couple of issues found with the manuscript. These have been addressed as outlined in our letter (attached).
